# Apoptosis and G2/M Phase Cell Cycle Arrest Induced by Alkaloid Erythraline Isolated from *Erythrina velutina* in SiHa Cervical Cancer Cell

**DOI:** 10.3390/ijms26104627

**Published:** 2025-05-12

**Authors:** Cleine Aglacy Nunes Miranda, Amaxsell Thiago Barros de Souza, Ana Katarina Menezes da Cruz Soares, Emanuelly Bernardes-Oliveira, Hugo Alexandre Oliveira Rocha, Euzébio Guimarães Barbosa, Thais Guaratini, Norma Lucena-Silva, Ricardo Ney Cobucci, Raquel Brandt Giordani, Janaina Cristiana de Oliveira Crispim

**Affiliations:** 1Postgraduate Program in Development and Technological Innovation in Medicines, Federal University of Rio Grande do Norte, Natal 52171-900, RN, Brazil; cleinemiranda.bio@hotmail.com (C.A.N.M.); bio_natalrn@yahoo.com.br (E.B.-O.); 2Postgraduate Program in Sciences Applied to Women’s Health, Federal University of Rio Grande do Norte, Natal 59012-310, RN, Brazil; thiagoamaxsell@gmail.com (A.T.B.d.S.); ricardo.cobucci.737@ufrn.edu.br (R.N.C.); 3Department of Biochemistry, Federal University of Rio Grande do Norte, Natal 59064-741, RN, Brazil; anakaty2018@gmail.com (A.K.M.d.C.S.); hugo.rocha@ufrn.br (H.A.O.R.); 4Department of Pharmacy, Federal University of Rio Grande do Norte, Natal 59012-570, RN, Brazil; euzebiogb@gmail.com (E.G.B.); raquel.giordani@ufrn.br (R.B.G.); 5Lychnoflora Research and Development in Natural Products LTDA, Ribeirão Preto 14040-900, SP, Brazil; thais@lychnoflora.com.br; 6Department of Immunology, Aggeu Magalhães Institute, Oswaldo Cruz Foundation, Recife 50670-465, PE, Brazil; norma.silva@fiocruz.br; 7Postgraduate Program in Biotechnology, Potiguar University, Natal 59056-000, RN, Brazil

**Keywords:** cervical cancer, erythraline, SiHa cells, apoptosis, cell cycle arrest

## Abstract

Cervical cancer remains a significant global health concern, causing more than 300,000 deaths annually. *Erythrina velutina*, a tree native to north-eastern Brazil, contains bioactive alkaloids with potential anticancer properties. This study aimed to characterize the alkaloid-enriched fraction of *Erythrina velutina* leaves and investigate the effects of the alkaloid erythraline on apoptosis and cell cycle in SiHa cervical cancer cells. Using Gas Chromatography–Mass Spectrometry (GC-MS), six alkaloids, including erythraline, were identified. Cytotoxicity was assessed through proliferation assays on SiHa cells and peripheral blood mononuclear cells (PBMCs). Apoptosis and cell cycle analyses were performed using flow cytometry, and in silico virtual screening identified potential protein targets of erythraline. Erythraline showed time- and concentration-dependent inhibitory effects on SiHa cell proliferation, with significant cytotoxicity observed at 50 µg/mL. Morphological changes, chromatin condensation, and increased apoptotic cell percentages confirmed the induction of caspase-independent apoptosis. Erythraline also induced G2/M cell cycle arrest, with 22% of cells in the G2/M phase compared with 7.25% in the untreated controls. In silico analysis identified polyamine oxidase, pyruvate kinase M2, and tankyrase as potential targets that contribute to the antitumor activity of erythraline. These findings suggest that erythraline is a promising candidate for anticancer therapy, warranting further investigation.

## 1. Introduction

Cervical cancer is the fourth most prevalent malignancy among women, resulting in over 300,000 deaths annually on a global scale [1]. Persistent infection with high-risk human papillomavirus (HPV) types, particularly HPV-16 and HPV-18, is the primary etiology of this disease [2]. Although prophylactic HPV vaccines have proven effective in preventing cervical cancer, they do not possess therapeutic properties and are unable to eliminate preexisting infections [3]. Current treatment modalities such as surgical intervention, chemotherapy, and radiotherapy often require combinatorial strategies and exhibit limited efficacy in numerous cases [4]. This highlights the pressing need for innovative and effective therapeutic alternatives.

*Erythrina velutina* Willd., commonly referred to as “mulungu”, is a pioneer tree endemic to the Caatinga, a tropical dry forest located in the semiarid northeastern region of Brazil. Species within the Erythrina genus are rich in bioactive metabolites, including alkaloids and flavonoids, which are of considerable pharmaceutical interest [5,6]. These bioactive compounds have been documented to exhibit antioxidant, anticonvulsant, and neuroprotective properties [7,8]. Moreover, recent investigations have demonstrated that a chymotrypsin inhibitor isolated from the seeds of *Erythrina velutina* can induce apoptosis and cell cycle arrest, thereby highlighting its potential as an anticancer agent [9].

Alkaloids, a diverse class of specialized organic nitrogenous bases derived from amino acids, have been recognized for their substantial antitumor properties. In plant systems, alkaloids function as protective agents against pathogens, herbivores, and abiotic stresses, including extreme fluctuations in light, water availability, and temperature [10]. Numerous alkaloids exhibit antitumor activity against various cancer types, including cervical cancer [11,12]. These characteristics render alkaloids as promising candidates for the development of novel anticancer therapies.

We hypothesized that alkaloids derived from *Erythrina velutina* would exhibit antitumor activity. To date, this hypothesis has not been investigated. Consequently, this study aimed to characterize the alkaloid-enriched fraction of *Erythrina velutina* and examine the apoptotic effects of the alkaloid erythraline, as well as its impact on the cell cycle in SiHa cervical cancer cells.

## 2. Results

### 2.1. Chemical Characterization of the Alkaloid-Enriched Fraction from Erythrina velutina Leaves

The compounds present in the alkaloid-enriched fraction of *Erythrina velutina* leaves were successfully characterized using Gas Chromatography–Mass Spectrometry (GC-MS) (Table 1). Six alkaloids were identified: erythrinine, erythraline, erythratidine, erythroculin, crystamidine, and 8-oxoerythraline (Figure 1). Notably, with the exception of erythraline and 8-oxoerythraline, the remaining alkaloids identified in this analysis have not been previously reported in *Erythrina velutina*.

### 2.2. Cytotoxicity

Erythraline exhibited an inhibitory effect on the proliferation of SiHa cells in a time- and concentration-dependent manner at concentrations of 50, 100, and 200 μg/mL. At the lowest concentration, erythraline was able to decrease cancer cell proliferation by over 50% within the initial 24 h. The determined IC50 value for erythraline was 35.25 μg/mL (~12 μM), which is notably lower than the reported IC50 value for cisplatin (17 μM).

To assess the cytotoxic effects on non-cancerous cells, peripheral blood mononuclear cells (PBMCs) were exposed to the same concentrations of erythraline. A significant reduction in PBMC viability was observed only at the highest concentration tested, in comparison to cisplatin (*p* < 0.05; Figure 2B).

Morphological changes in SiHa cells were analyzed by optical microscopy and compared with those in untreated controls. Untreated SiHa cells displayed well-adhered growth, a fusiform shape, a high refractive index, and clear cytoplasm (Figure 3). Following erythraline treatment (100 and 200 μg/mL) for 48 h, the cells exhibited rounding (black arrow), decreased size, and detachment from the monolayer, indicating a marked reduction in the cell number.

To investigate chromatin alterations, cells were stained with 4′,6-diamidino-2-phenylindole (DAPI) following a 48 h treatment period. Untreated cells exhibited a rounded morphology with uniformly stained nuclei (Figure 4A). In contrast, cells treated with 50 μg/mL erythraline demonstrated nuclear retraction and chromatin condensation (Figure 4C). Similarly, cells treated with cisplatin exhibited characteristics such as pyknotic nuclei, chromatin condensation, and nuclear fragmentation (Figure 4B).

### 2.3. Erythraline Induces Apoptosis in SiHa Cells

The apoptotic effect of erythraline on SiHa cells was evaluated utilizing Annexin V-fluorescein isothiocyanate (FITC) and propidium iodide (PI) staining. Following a 48 h treatment, the percentage of apoptotic cells (AV+/PI− and AV+/PI+) exhibited a dose-dependent increase, reaching a peak of 87.9% at a concentration of 50 μg/mL (Figure 5). A significant proportion of the cells were identified as being in the late apoptotic phase (AV+/PI+), indicating effective induction of apoptosis by erythraline. Furthermore, treatment with erythraline at 50 μg/mL resulted in a significantly greater reduction in viable cells (1.35%) in comparison to cisplatin (14.1%).

### 2.4. Caspase Activity Investigation

The role of caspases in erythraline-induced cell death was investigated. SiHa cells were treated with erythraline (50 μg/mL) in the presence of 100 μM z-VAD, a caspase inhibitor. Despite the z-VAD treatment, a quantitative increase in cell death was observed (Figure 6). As apoptotic processes are generally mediated by caspase activation, the lack of caspase inhibition suggests that erythraline-induced SiHa cell death is caspase-independent.

### 2.5. Erythraline Increases G2/M Arrest of Cell Cycle in SiHa Cells

Flow cytometry was used to analyze the effect of erythraline on the cell cycle after 24 h of treatment (Figure 7). Erythraline-treated SiHa cells (50 μg/mL) exhibited an increased proportion of cells in the G2/M phase (22%) compared with untreated controls (7.25%). This was accompanied by a decrease in G1-phase cells. These findings suggest that erythraline-induced growth inhibition and apoptosis of SiHa cells are partially associated with G2/M cell cycle arrest.

### 2.6. Analysis of Mitochondrial Membrane Potential with Rhodamine 123

Mitochondrial dysfunction is a characteristic feature of apoptotic cell death. To investigate this phenomenon, the disruption of mitochondrial membrane potential (ΔΨm) was assessed in SiHa cells treated with erythraline. Changes in ΔΨm were evaluated using the mitochondria-specific dye Rhodamine 123 (Rh-123). As illustrated in Figure 8, treatment with 100 μg/mL erythraline for 24 h, followed by Rh-123 staining, did not yield significant alterations in ΔΨm when compared to the control group. These findings suggest that erythraline treatment in SiHa cells does not induce a collapse of the mitochondrial membrane potential. In contrast, cisplatin treatment resulted in slight mitochondrial membrane depolarization after 24 h of exposure.

### 2.7. In Silico Virtual Screening of Erythraline and Antitumor Targets

Virtual screening identified several potential protein targets of erythraline. Among the top 25 ligands with high similarity scores, Polyamine Oxidase (PAO), Pyruvate Kinase M2 (PKM2), and tankyrase (TANK) emerged as potential candidates for the antitumor activity of erythraline (Figure 9).

## 3. Discussion

This study successfully characterized an alkaloid-enriched fraction derived from the leaves of *Erythrina velutina* and identified six distinct alkaloids, including erythraline and 8-oxoerythraline. Notably, the remaining four alkaloids have not previously been documented in this species. Erythraline, which served as the primary focus of this investigation, exhibited potent cytotoxic effects against SiHa cervical cancer cells, prompting apoptosis and G2/M cell cycle arrest.

Multiple studies have reported the cytotoxic potential of Erythrina species including *Erythrina velutina* [17,18]. However, the specific alkaloids implicated in these effects and their underlying mechanisms of action remain poorly understood. Our findings highlight the potential of erythraline as a promising anticancer agent. The calculated IC50 value for erythraline was determined to be 35.25 μg/mL (12 μM). Notably, this IC50 value is lower than that reported for cisplatin (17 μM) under similar experimental conditions [19], a well-established chemotherapeutic agent. This comparison suggests that erythraline exhibits comparable cytotoxicity to cisplatin, underscoring its potential as a promising therapeutic candidate in the treatment of cervical cancer. For instance, a study conducted by Mohammed et al. demonstrated the cytotoxic activity of erythraline against various Hep-G2 cancer cell lines [17]. Similarly, Armwood et al. reported the apoptotic effects of erythraline, associated with the activation of caspase-3 [18].

As previously demonstrated in lung cancer cell lines, pretreatment with ZVAD inhibited cisplatin- and staurosporine-induced apoptosis, indicating the occurrence of caspase-independent apoptosis in these cells. Similarly, our findings indicate that erythraline induces caspase-independent apoptosis. Recent studies on alkaloids have identified alternative pathways for caspase-independent cell death. For instance, a study on camptothecin-induced apoptosis reported nuclear apoptosis and mitochondrial dysfunction, highlighting potential mitochondrial involvement [20]. Furthermore, research on xylopine demonstrated its ability to induce oxidative stress and arrest the cell cycle at the G2/M phase, triggering caspase-mediated apoptosis through a p53-independent pathway in HCT116 cells [21].

In contrast, our results suggest that mitochondrial pathways are not involved in erythraline-induced apoptosis, as no changes in ΔΨm were observed. This observation aligns with a previous study, which showed that ΔΨm reduction in Jurkat cells exposed to Ca2+ ionophores occurs only in the presence of ZVAD [22]. These findings imply that erythraline induces apoptosis through alternative mechanisms, potentially involving endonucleases or other non-mitochondrial proteins. Further investigation is required to elucidate the precise molecular pathways underlying erythraline-induced apoptosis.

The identification of potential protein targets of erythraline, such as PAO, PKM2, and TANK, provides valuable insights into its mechanism of action. PAO is involved in polyamine metabolism and is frequently dysregulated in cancer cells. Inhibition of PAO has been shown to induce apoptosis and inhibit tumor growth, as evidenced in a previous study [23]. PKM2 plays a critical role in glycolysis and is often overexpressed in cancer cells. Targeting of PKM2 has emerged as a promising anticancer strategy [24,25,26]. TANK, a member of the poly (ADP-ribose) polymerase (PARP) family, is involved in various cellular processes including DNA repair and cell proliferation. Inhibition of TANK has been demonstrated to induce apoptosis and enhance the sensitivity of head and neck squamous cells to chemotherapy [27,28].

While this study provides valuable insights into the cytotoxic potential of erythraline and its prospective molecular targets, further research is warranted to deepen our understanding. Future investigations should focus on elucidating the specific mechanisms underlying erythraline-induced apoptosis and cell cycle arrest, with particular attention to its effects on key signaling pathways and gene expression. Moreover, a significant limitation of this study is the absence of an evaluation of erythraline’s antiviral activity against HPV. Ultimately, in vivo studies are crucial for determining the therapeutic efficacy of erythraline and assessing its potential toxic effects.

## 4. Materials and Methods

### 4.1. Plant Material

Leaves of Erythrina velutina were collected in March 2013 during the dry season in Acari, Rio Grande do Norte, Brazil (06°26′08″ S, 36°38′20″ W). A voucher specimen was deposited in the Herbarium of the Federal University of Rio Grande do Norte under reference number 21666. Authorization for harvesting plant material was granted by SISBIO (Permit No. 32749-2), and access to Brazil’s genetic patrimony was approved by CNPq (Permit No. 010142/2012-6).

### 4.2. Extraction and Isolation

Fresh leaves (478 g) were dried in an air-circulating oven at 45 °C, crushed, and extracted thrice with 96% ethanol (1:10 *w*/*v*) at room temperature for 24 h. The resulting crude extract was concentrated under reduced pressure, acidified with 10% HCl, and subjected to acid–base extraction with n-hexane at pH 2. The extract was basified to a pH 10 using NH_4_OH and re-extracted with chloroform. After evaporation under reduced pressure, the alkaloid-enriched fraction obtained from the chloroform at pH 10 was isolated.

GC-MS analysis of the alkaloid-enriched fraction was conducted as previously described. The samples were injected in split less mode at 250 °C using helium as the carrier gas at a flow rate of 1.1 mL/min. The temperature gradient was as follows: 100–220 °C at 6 °C/min (held for 10 min), followed by 220–290 °C at 6 °C/min (held for 11 min). The injection volume was 1 μL and the samples were solubilized in methanol. Mass spectra were acquired via electron ionization at 70 eV. The alkaloid erythraline was provided by Lychnoflora LTDA (Ribeirão Preto, SP, Brazil).

### 4.3. Reagents and Antibodies

DAPI, 3-(4,5-dimethylthiazol-2-yl)-2,5-diphenyltetrazolium bromide (MTT), sodium pyruvate, essential amino acids, trypsin, and dimethyl sulfoxide (DMSO) were purchased from Sigma-Aldrich (Louis, MO, USA). Dulbecco’s modified Eagle’s medium (DMEM) and fetal bovine serum (FBS) were obtained from Cultilab (Campinas, SP, Brazil). Annexin V-FITC/PI for flow cytometry were obtained from Invitrogen (Waltham, MA, USA), and CDDP (50 mg) was acquired from Bergamo (Taboão da Serra, SP, Brazil). z-VAD was generously provided by Professor Dr. Aurigena Antunes de Araújo from the Department of Biophysics and Pharmacology at the Federal University of Rio Grande do Norte.

### 4.4. Cell Lines and Culture

SiHa cells were cultured and made available at the Cell Culture Laboratory of the Biochemistry Department of the Federal University of Rio Grande do Norte (UFRN). SiHa cells (HPV16 and p53wt) were maintained in complete DMEM, whereas PBMC were cultured in RPMI 1640 medium. Both media were supplemented with 10% fetal bovine serum (FBS), 100 U/mL penicillin, and 100 µg/mL streptomycin. Lymphocytes were isolated from human peripheral blood using Ficoll-Paque from Amersham Pharmacia Biotech (Upsalla, Sweden) according to the manufacturer’s instructions. All the cells were incubated at 37 °C in a humidified atmosphere containing 5% CO_2_. Cells were seeded overnight and treated with various concentrations of erythraline for specific durations. CDDP was used as the positive control.

### 4.5. Cell Viability Assay

Cells were seeded in 96-well plates at a density of 5000 cells/well in 100 µL of complete medium and subsequently treated with erythraline for 24 or 48 h. Following treatment, 10 µL of MTT solution (5 mg/mL) was added to each well and the plates were incubated for 4 h. The resulting formazan crystals were dissolved in 100 µL of DMSO, and the absorbance was measured at 570 nm, with 620 nm serving as a reference wavelength, using an ELISA reader (BioTek Instruments, Winooski, VT, USA). Cell viability was expressed as a percentage relative to the vehicle-treated controls. Each assay included a positive control of cisplatin at concentrations of 15 or 30 μg/mL and a negative control of 1% DMSO. DMSO was used as the negative control because it served as the solvent for the alkaloid fractions.

### 4.6. Fluorescence Microscopy

Cells were cultivated on coverslips in 24-well plates at a density of 30,000 cells/well and treated with erythraline (100 and 200 µg/mL) and cisplatin 30 μg/mL for 48 h. Following treatment, the cells were washed with phosphate-buffered saline (PBS), fixed in 4% paraformaldehyde for 30 min, and permeabilized with 0.1% Triton X-100 at room temperature for an additional 30 min. After further washing with PBS, the cells were stained with DAPI (5 µg/mL) for 30 min, washed again, and visualized using the Eclipse Ti-U fluorescence microscope (Nikon Instruments Inc., Tokyo, Japan).

### 4.7. Apoptosis Analysis

Apoptosis in SiHa cells was assessed using Annexin V-FITC/PI staining combined with flow cytometry. The cells were treated for 48 h, harvested, and washed twice with cold PBS. They were then incubated with 5 µL of Annexin V and binding buffer for 10 min in the dark at room temperature, followed by the addition of 1 µL of PI for 5 min, according to the manufacturer’s instructions (Invitrogen, Catalog No. V13242). Stained cells were analyzed using a FACS Calibur flow cytometer (Becton Dickinson, Franklin Lakes, NJ, USA), and apoptotic rates were calculated as the percentage of apoptotic cells among the total population. Inhibition of apoptosis was evaluated using 100 µM z-VAD-FMK. The assay was conducted with a positive control (cisplatin, 30 μg/mL) and a negative control (1% DMSO).

### 4.8. Cell Cycle Analysis

Cell cycle distribution was analyzed by PI staining. Cells (1 × 10^6^) were seeded in 6-well plates, treated for 24 h, harvested, and fixed with 70% ethanol at −20 °C overnight. After two washes with PBS, the cells were incubated with RNase A (1 mg/mL) for 15 min and then stained with 10 µg/mL PI for 10 min in the dark at room temperature. DNA content was analyzed using flow cytometry with FACS Calibur and FlowJo software (version X10.0.7, Tree Star, Inc., Ashland, OR, USA). Untreated cells were used as the control.

### 4.9. Rhodamine-123 Retention Assay

The retention of Rhodamine 123 (Rh-123) was evaluated using flow cytometry as a functional indicator of P-glycoprotein (P-gp) activity. A total of 1 × 10^5^ cells were treated with erythraline 100 μg/mL and cisplatin 15 μg/mL for 24 h prior to the addition of Rh-123 at a concentration of 10 mg/L. Following a 1 h incubation at 37 °C, the cells were harvested and subjected to centrifugation for 10 min. The resulting cell pellets were resuspended in 500 μL of PBS and subsequently analyzed for Rh-123 retention using flow cytometry.

### 4.10. Molecular Protein–Protein Docking

The three-dimensional structure of erythraline was generated using Avogadro software (version 1.2.0) and subsequently optimized using PM7 semi-empirical theory within the MOPAC software (version 2016). The optimized structure was screened against a database of over 9000 proteins from the Protein Data Bank PDB www.rcsb.org) to identify potential targets. Molecular docking was conducted using AutoDock Vina (version 1.1.2), and the binding modes were visualized using UCSF Chimera software (version 1.12).

### 4.11. Statistical Analysis

Data were analyzed using GraphPad InStat^®^ (version 4.0; GraphPad Software, San Diego, CA, USA). Results are presented as mean ± standard deviation (SD). One-way ANOVA followed by Tukey’s multiple comparison test was applied. Statistical significance was defined as *p* < 0.05.

## 5. Conclusions

Erythraline, the alkaloid-enriched fraction of *Erythrina velutina*, exhibited potent cytotoxic effects against SiHa cervical cancer cells, prompting apoptosis and G2/M cell cycle arrest. The identification of erythraline as a potent cytotoxic agent with potential antitumor activity underscores the need for further investigation of its development as a novel anticancer drug.

## Figures and Tables

**Figure 1 ijms-26-04627-f001:**
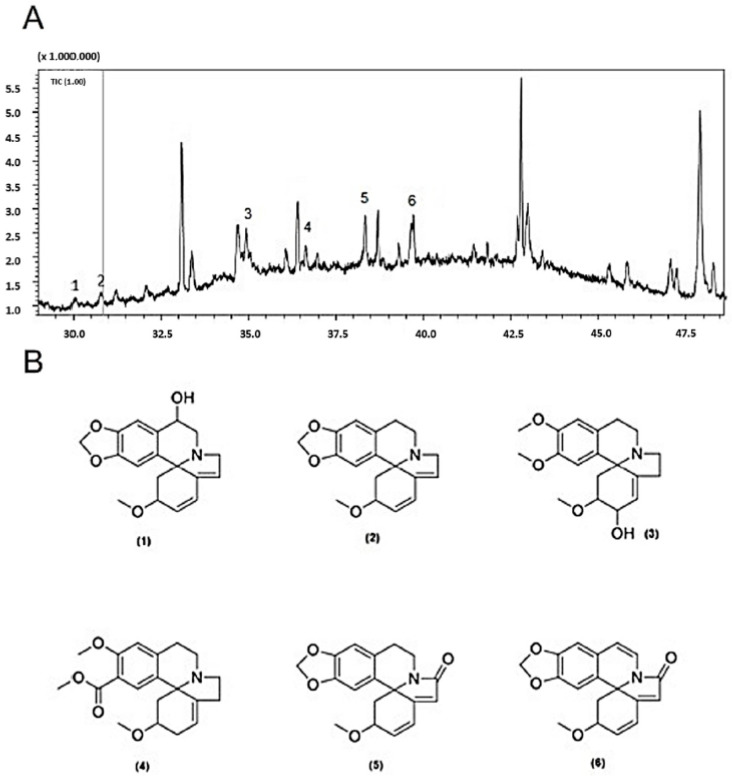
(**A**) Amplification of the chromatogram (CG-MS) of *E. velutina* extract. (**B**) 2D structures of the alkaloids identified by CG-MS in the *E. velutina* extract sample. (**1**): Erythrinine; (**2**): Erythraline; (**3**): Erythratidine; (**4**): Erythroculin; (**5**): Cristamidine; (**6**): Oxoerythraline.

**Figure 2 ijms-26-04627-f002:**
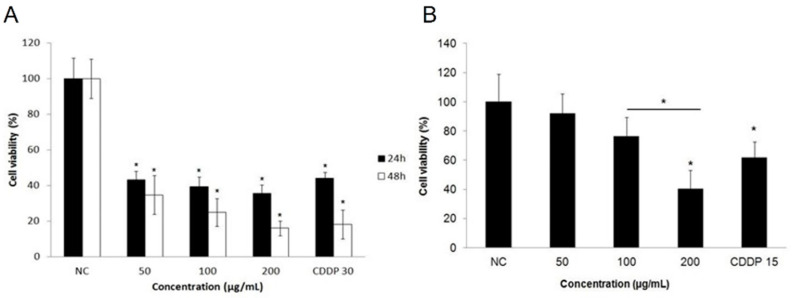
(**A**) Cells were treated with 50, 100, and 200 μg/mL erythraline and cisplatin (CDDP) for 24 h and 48 h and in (**B**) PBMC cells for 24 h. NC, control. * *p* < 0.05.

**Figure 3 ijms-26-04627-f003:**
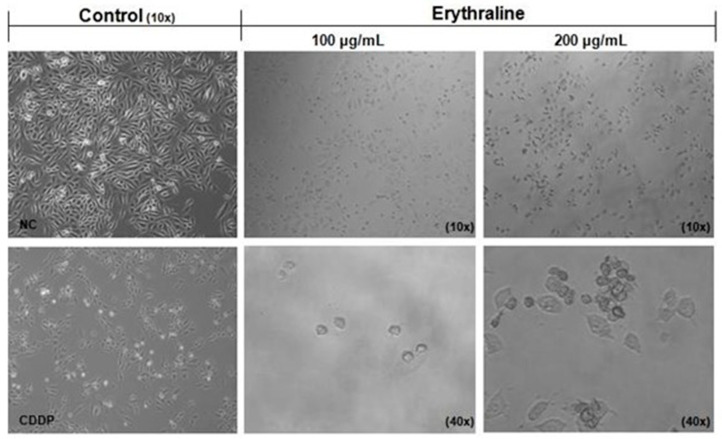
Morphological changes in the SiHa cell after treatment with erythraline for 48 h.

**Figure 4 ijms-26-04627-f004:**
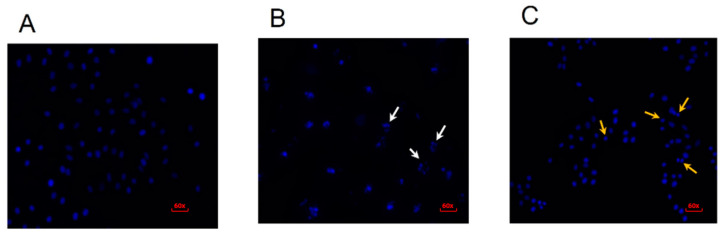
Fluorescence microscopy of cells treated with erythraline and stained with DAPI after 48 h of treatment in (**A**) control, (**B**) cisplatin 30 μg/mL, and (**C**) erythraline 50 μg/mL. White arrows indicate nuclear fragmentation and apoptotic bodies. Yellow arrows indicate nuclear retraction and chromatin condensation.

**Figure 5 ijms-26-04627-f005:**
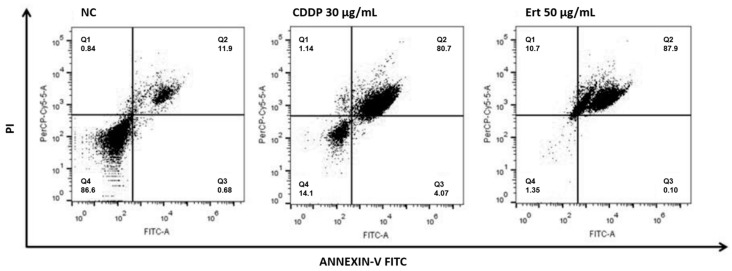
Flow cytometry analysis of SiHa cells treated with Ert (erythraline) for 48 h compared to NC (control) and CCDP (cisplatin).

**Figure 6 ijms-26-04627-f006:**
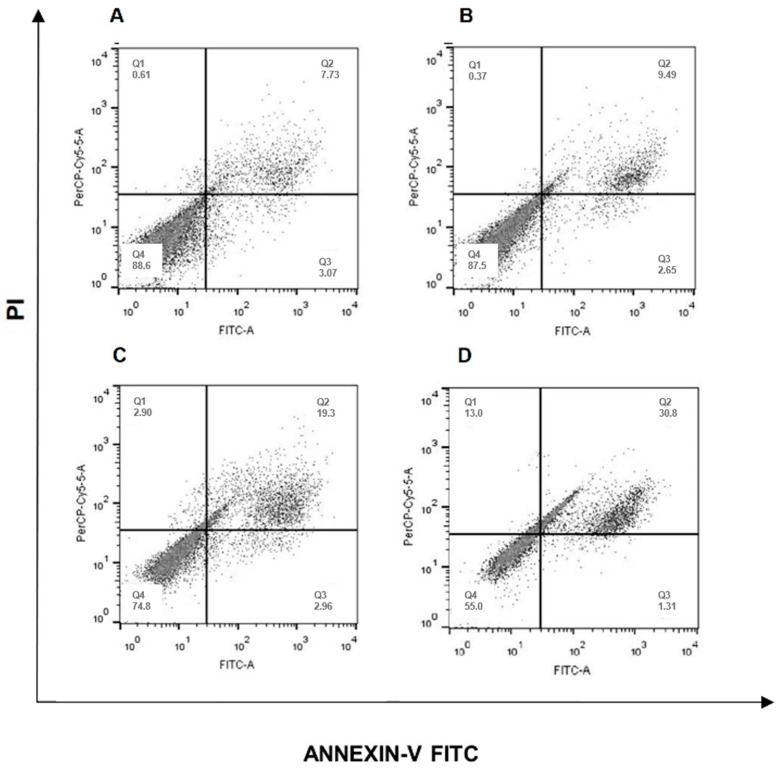
Flow cytometric of apoptosis analysis of SiHa cells treated with isolated erythraline or associated with z-VAD (100 μM). In (**A**) untreated cells; (**B**) cells with z-VAD (100 μM) addition; in (**C**) cells treated with erythraline; and in (**D**) treatment with the erythraline associated with z-VAD.

**Figure 7 ijms-26-04627-f007:**
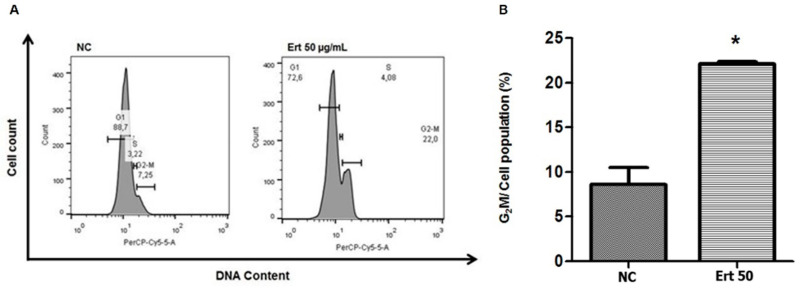
(**A**) Flow cytometric of cell cycle analysis of SiHa cells treated with isolated erythraline (Ert) or control (NC); (**B**) comparison of G2/M cell count in Ert-treated group and NC. * *p* < 0.05.

**Figure 8 ijms-26-04627-f008:**
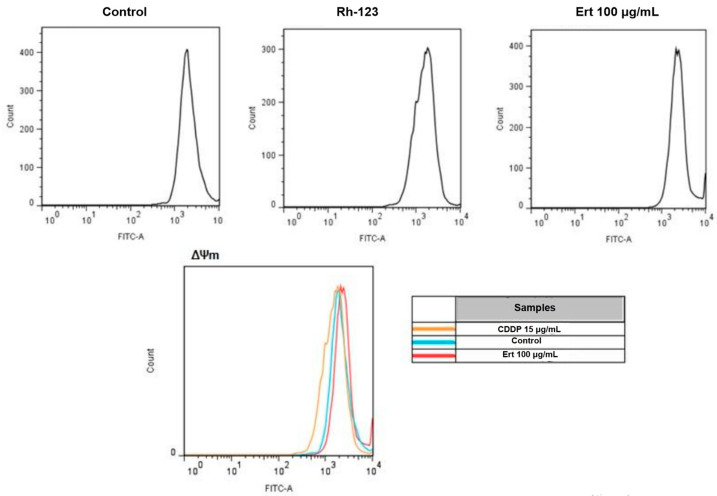
Analysis of mitochondrial membrane potential in SiHa cells after treatment with erythraline (Ert) 100 μg/mL, cisplatin (CDDP) 15 μg/mL, and control for 24 h. The mitochondrial membrane potential was measured using Rhodamine 123 (Rh-123) and analyzed by flow cytometry.

**Figure 9 ijms-26-04627-f009:**
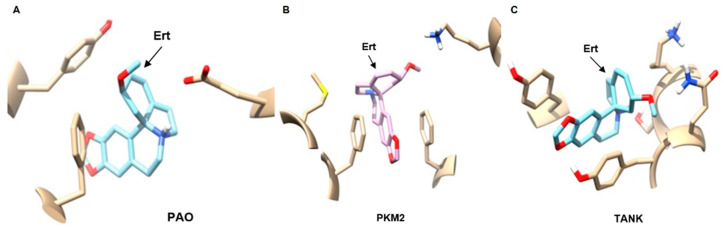
Computational model of the target binding interaction of the erythraline alkaloid (Ert) and the proteins (**A**) Polyamine Oxidation (PAO), (**B**) Pyruvate Kinase M2 (PKM2), and (**C**) Tankyrase (TANK).

**Table 1 ijms-26-04627-t001:** GC-MS data of erythrina alkaloids identified in *Erythrina velutina* extract.

Peak	Tr (min)	Alkaloid	*m*/*z* (IE-MS)	*m*/*z* (IE-MS) Literature
1	30	Erythrinin	295 (38), 280 (49), 264 (100), 262 (24), 224 (23), 211 (21)	295 (32), 280 (37), 265 (26), 264 (100), 262 (19), 224 (12), 211 (28) [13]
2	30.8	Erythraline	297 (29), 282 (14), 267 (17), 266 (100), 226 (20), 213 (21)	297 (35), 282 (31), 267 (26), 266 (100), 264 (21), 239 (11), 226 (13), 213 (22) [13]
3	34.9	Erythratidine	331 (3), 300 (20), 273 (83), 272 (37), 258 (32), 257 (100), 256 (34), 244 (84)	331, 300, 273, 257 (100), 244b, 331 (5), 300 (13), 273 (50), 272 (15), 258 (27), 257 (100), 256 (54), 244 (60), 242 (15) [14]
4	36.4	Erythroculin	343 (80), 328 (21), 286 (66), 285 (100), 271 (24), 256 (37), 255 (71)	343 (71), 328 (0.5), 312 (13), 285 (100) [15]
5	36.6	Crystamidine	309 (30), 294 (21), 278 (39), 277 (40), 276 (100), 266 (26), 250 (57)	309, 294, 278, 276 (100) [16]
6	39.7	Oxo-erythraline	311 (74), 296 (40), 280 (45), 279 (59), 278 (100), 268 (13), 251 (26), 250 (54)	311 (81), 296 (42), 280 (57), 279 (51), 278 (100), 268 (20), 266 (17), 250 (48) [13]

## Data Availability

The raw data supporting the conclusions of this article will be made available by the authors on request.

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
