# Peer review of "Apoptosis and G2/M Phase Cell Cycle Arrest Induced by Alkaloid Erythraline Isolated from *Erythrina velutina* in SiHa Cervical Cancer Cell"

_ijms, 2025, doi:10.3390/ijms26104627_

Round 1

Reviewer 1 Report

Comments and Suggestions for Authors

Cervical cancer is a common malignancy in women and constitutes a serious public health problem worldwide.  Persistent infection with high-risk human papillomaviruses (HPV), especially HPV-16 and HPV-18, plays an important role in the etiology of this disease. 

The Authors decided to check whether alkaloids derived from Erythrina velutina would have anticancer effects.  For this purpose, the Erythrina velutina fraction was first characterized and then the effects of the apoptotic alkaloid erythraline were examined, as well as its impact on the cell cycle in SiHa cervical cancer cells. 

The studies were carried out in cell lines, first assessing cytotoxicity using commonly used methods. Apoptosis was assessed by Annexin V-129-cein isothiocyanate (FITC) and propidium iodide (PI) staining. 

Erythraline had a strong cytotoxic effect on SiHa cervical cancer cells, causing apoptosis and G2/M cell cycle arrest. I believe that these are preliminary studies on erythraline as a cytotoxic agent with potential anticancer activity and require further in-depth research on its possible usefulness in development as a potential anticancer drug. 

Since the Authors write that cervical cancer is caused by HPV viruses, it should be tested whether this extract has any antiviral activity. Otherwise, it seems that such a study is not complete. Either the Authors will supplement them with additional research or they will comment on this aspect in the discussion or the limitations of the research.

Author Response

Comment 1: Cervical cancer is a prevalent malignancy in women, constituting a significant public health concern worldwide. Persistent infection with high-risk human papillomaviruses (HPV), particularly HPV-16 and HPV-18, plays a critical role in the etiology of this disease. The authors investigated the potential anticancer effects of alkaloids derived from Erythrina velutina. This included characterizing the Erythrina velutina fraction and evaluating the cytotoxic effects of erythraline, particularly its induction of apoptosis and G2/M cell cycle arrest in SiHa cervical cancer cells. The study provides a promising foundation; however, since cervical cancer is caused by HPV, it is essential to determine whether the extract also possesses antiviral activity. Without this, the study remains incomplete. The authors should either incorporate antiviral studies or address this limitation in the discussion.

Response 1: We fully agree with the reviewer’s observation. As recommended, we have acknowledged this limitation in the discussion section. The revised manuscript now includes the following statement: “Moreover, a significant limitation of this study is the absence of an evaluation of erythraline's antiviral activity against HPV. Ultimately, in vivo studies are crucial for determining the therapeutic efficacy of erythraline and assessing its potential toxic effects.” (lines 249 - 251).

Reviewer 2 Report

Comments and Suggestions for Authors

This manuscript by Cleine Aglacy Nunes Miranda et al., titled Apoptosis and G2/M Phase Cell Cycle Arrest Induced by Alkaloid Erythraline Isolated from Erythrina velutina in SiHa Cervical Cancer Cell, presents a praiseworthy findings of the experimental rigor and pharmacological and mechanistic aspects of Erythraline alkaloids. While the authors’ efforts are exemplary, a closer evaluation highlights areas that would benefit from additional study and clarification. Addressing these points could greatly enhance the scientific impact of this work and support its well-deserved acceptance.

Detailed Comments:

1.     The study does not sufficiently justify focusing solely on erythraline while excluding other isolated alkaloids, such as erythrinine, erythratidine, erythroculin, crystamidine, and 8-oxoerythraline, despite noting their novelty in the results section. Providing a clear rationale for this choice would strengthen the study's foundation. Including these alkaloids in the experimental analysis to conduct a comprehensive screening profile, and SAR analysis could significantly enhance the therapeutic understanding of erythraline and address potential limitations. In addition, understanding how structural variations influence their differential modulatory effects, selectivity, potency and safety profile, could provide valuable insights for optimizing their pharmacological profiles and advancing drug design.

2.     While the study demonstrates erythraline’s cytotoxicity against SiHa cells, it does not provide an IC50 value comparison with standard cervical cancer drugs, such as cisplatin or paclitaxel. Incorporating such comparisons would contextualize erythraline’s potency and therapeutic potential, establishing its relative efficacy more clearly.

3.     The study suggests erythraline induces caspase-independent apoptosis but does not delve into the underlying mechanisms. A detailed exploration of specific pathways or molecules involved in this process, such as mitochondrial pathways, autophagy markers, or other apoptosis-related proteins, would offer deeper mechanistic insights and enhance the robustness of the findings.

4.     Although erythraline’s cytotoxicity was evaluated in PBMCs, additional data on its selectivity and off-target effects are necessary to assess its therapeutic window and safety profile. Expanding the cytotoxicity analysis to include non-cancerous epithelial cells, such as HaCaT or normal cervical cells, would provide a clearer picture of erythraline’s selectivity. Furthermore, determining its therapeutic index would emphasize its safety margin and clinical viability.

5.     The study does not investigate whether erythraline could enhance or synergize with existing cervical cancer treatments. Exploring this aspect could reveal combination strategies that improve therapeutic outcomes and broaden erythraline’s clinical applications.

Author Response

Comment 1: The study lacks justification for focusing solely on erythraline while excluding other isolated alkaloids, such as erythrinine, erythratidine, erythroculin, crystamidine, and 8-oxoerythraline, despite their novelty. A rationale for prioritizing erythraline would strengthen the study’s foundation. Additionally, including these alkaloids in a comprehensive screening profile or structure-activity relationship (SAR) analysis could enhance the understanding of their pharmacological potential.

Response 1: We appreciate the reviewer’s suggestion. While we fully acknowledge the importance of a broader analysis, we regret that our current funding constraints prevent us from conducting experiments on additional alkaloids.

Comment 2: The study demonstrates erythraline’s cytotoxicity in SiHa cells but does not compare its IC50 value to standard cervical cancer drugs like cisplatin or paclitaxel. Such comparisons would contextualize erythraline’s therapeutic potential.

Response 2: Thank you for this insightful comment. In response, we have incorporated IC50 value comparisons into the manuscript. The results section now states: “At the lowest concentration, erythraline was able to reduce cancer cell proliferation by over 50% within the first 24 hours. The calculated IC50 value for erythraline was 35.25 μg/mL (~12 μM). Notably, this IC50 value is lower than that reported for cisplatin (17 μM). To evaluate the cytotoxic effects on non-cancerous cells, peripheral blood mononuclear cells (PBMCs) were subjected to the same concentrations of erythraline. A significant reduction in PBMC viability was noted only at the highest concentration tested, relative to cisplatin (p < 0.05; Figure 2B).” (lines 96-102).

Similarly, the discussion has been updated to emphasize this comparison: “Our findings highlight the potential of erythraline as a promising anticancer agent. The calculated IC50 value for erythraline was 35.25 μg/mL (~12 μM). Notably, this IC50 value is lower than that reported for cisplatin (17 μM) under similar experimental conditions [15], a well-established chemotherapeutic agent. This comparison suggests that erythraline exhibits comparable cytotoxicity to cisplatin, highlighting its potential as a therapeutic candidate for cervical cancer treatment.” (lines 208-213).

Comment 3: While the study suggests that erythraline induces caspase-independent apoptosis, it does not investigate the underlying mechanisms. A detailed exploration of specific pathways or molecules involved in this process would enhance the study’s robustness.

Response 3: We appreciate this important suggestion. While limited funding precludes us from conducting extensive pathway analyses, we included an experiment evaluating mitochondrial membrane potential to provide additional context. This has been incorporated into the methods (section 4.9), results (section 2.6), and discussion sections. The discussion now includes: “In contrast, our results suggest that mitochondrial pathways are not involved in erythraline-induced apoptosis, as no changes in ΔΨm were observed. This observation aligns with a previous study, which showed that ΔΨm reduction in Jurkat cells exposed to Ca2+ ionophores occurs only in the presence of ZVAD [22]. These findings imply that erythraline induces apoptosis through alternative mechanisms, potentially involving endonucleases or other non-mitochondrial proteins. Further investigation is required to elucidate the precise molecular pathways underlying erythraline-induced apoptosis.” (lines 227-233).

Comment 4: Additional data on erythraline’s selectivity and off-target effects are required to assess its therapeutic index. Evaluating its cytotoxicity in non-cancerous epithelial cells, such as HaCaT or normal cervical cells, would provide clearer insights.

Response 4: We agree with the reviewer’s assessment. Unfortunately, due to resource limitations, we could not acquire non-tumor control cells. However, we conducted experiments using PBMCs from healthy patients to evaluate erythraline’s selectivity.

Comment 5: The study does not explore whether erythraline could synergize with existing cervical cancer treatments. Investigating combination strategies could broaden its clinical applications.

Response 5: Thank you for this valuable comment. While the study did not explicitly examine synergistic effects, we conducted cytotoxicity experiments and apoptosis assays in comparison to cisplatin. These data provide preliminary insights into erythraline’s potential applications alongside established treatments.

Round 2

Reviewer 1 Report

Comments and Suggestions for Authors

The authors made corrections both in Materials and Methods and in the Discussion. This undoubtedly improved the readability and transparency of the text.